# Effects of Paired Associative Stimulation on Cortical Plasticity in Agonist–Antagonist Muscle Representations

**DOI:** 10.3390/brainsci13030475

**Published:** 2023-03-10

**Authors:** Makoto Suzuki, Kazuo Saito, Yusuke Maeda, Kilchoon Cho, Naoki Iso, Takuhiro Okabe, Takako Suzuki, Junichi Yamamoto

**Affiliations:** 1Faculty of Health Sciences, Tokyo Kasei University, 2-15-1 Inariyama, Sayama City 350-1398, Saitama, Japan; 2Faculty of Systems Design, Tokyo Metropolitan University, 1-1 Minami-Osawa, Hachioji City 192-0397, Tokyo, Japan; 3School of Health Sciences at Odawara, International University of Health and Welfare, 1-2-25 Shiroyama, Odawara City 250-8588, Kanagawa, Japan; 4School of Health Sciences, Saitama Prefectural University, 820 Sannomiya, Koshigaya City 343-8540, Saitama, Japan

**Keywords:** paired associative stimulation, long-term potentiation, agonist–antagonist muscle, magnetic stimulation, human

## Abstract

Paired associative stimulation (PAS) increases and decreases cortical excitability in primary motor cortex (M1) neurons, depending on the spike timing-dependent plasticity, i.e., long-term potentiation (LTP)- and long-term depression (LTD)-like plasticity, respectively. However, how PAS affects the cortical circuits for the agonist and antagonist muscles of M1 is unclear. Here, we investigated the changes in the LTP- and LTD-like plasticity for agonist and antagonist muscles during PAS: 200 pairs of 0.25-Hz peripheral electric stimulation of the right median nerve at the wrist, followed by a transcranial magnetic stimulation of the left M1 with an interstimulus interval of 25 ms (PAS-25 ms) and 10 ms (PAS-10 ms). The unconditioned motor evoked potential amplitudes of the agonist muscles were larger after PAS-25 ms than after PAS-10 ms, while those of the antagonist muscles were smaller after PAS-25 ms than after PAS-10 ms. The γ-aminobutyric acid A (GABA_A_)- and GABA_B_-mediated cortical inhibition for the agonist and antagonist muscles were higher after PAS-25 ms than after PAS-10 ms. The cortical excitability for the agonist and antagonist muscles reciprocally and topographically increased and decreased after PAS, respectively; however, GABA_A_ and GABA_B_-mediated cortical inhibitory functions for the agonist and antagonist muscles were less topographically decreased after PAS-10 ms. Thus, PAS-25 ms and PAS-10 ms differentially affect the LTP- and LTD-like plasticity in agonist and antagonist muscles.

## 1. Introduction

Functional interactions between the primary motor cortex (M1) neurons are critical for human movement. Paired associative stimulation (PAS) has been widely used in recent decades to induce synaptic plasticity in M1 neurons, in which electrical peripheral nerve stimulation is paired with M1 stimulation [1,2]. In spike timing-dependent plasticity, synaptic potentiation is increased (long-term potentiation [LTP]) if the presynaptic spikes precede postsynaptic action potentials [3], whereas synaptic potentiation is decreased (long-term depression [LTD]) if the postsynaptic spikes precede the presynaptic action potentials [4,5]. Previous studies [1,2,6,7] have suggested that PAS induces LTP- and LTD-like plasticity through spike timing-dependent plasticity. Specifically, if the peripheral input precedes M1 stimulation due to peripheral electrical nerve stimulation at 25 ms (PAS-25 ms) before the transcranial magnetic stimulation (TMS) over M1, PAS can result in an increased cortical excitability (LTP-like plasticity) [1,8,9]. In contrast, if the peripheral input follows M1 stimulation due to peripheral electrical nerve stimulation at 10 ms (PAS-10 ms) before the TMS over M1, cortical excitability is reduced (long-term depression (LTD)-like plasticity) [1,2]. Additionally, the synaptic plasticity induced by PAS is suggested to follow topographical rules [2,6,10], because the homologous connection of the somatosensory cortex and M1 has a high topographical specificity [11]. Previous studies [12,13,14,15] have also suggested that PAS induces the selective reinforcement of γ-aminobutyric acid B (GABA_B_)-ergic cortical circuits. The topography-specific LTP- and LTD-like plasticity of PAS persists after stimulation [9,16,17]. Therefore, PAS can be applied to patients with motor control deficits, such as stroke [18], Parkinson’s disease [19,20], and Huntington’s disease [21].

In motor control, layers III and V of M1 bidirectionally connect different muscle representations [22]. The output from the common M1 site could diverge onto agonist and antagonist muscles with different “gain” [17,23,24,25,26], and the stimulation of the M1 region could elicit particular movements [27,28]. Therefore, movements, including those of the agonist and antagonist muscles, could be controlled by a network within the M1 [22,23]. Specifically, the control of agonist–antagonist muscles is crucial for smooth and precise motor control [24,29,30]. Several studies have suggested that the motor representations of M1 can be reorganized during motor training [31,32,33,34,35] and that competition with antagonist muscle representations could occur by the expansion of the trained agonist muscle representation [31,36,37]. However, the changes in representations for the agonist and antagonist muscles after PAS remain unclear. Therefore, despite the fact that functional interaction in M1 is crucial for human movement, the changes induced by PAS in the cortical circuits for agonist–antagonist muscle representations of M1 remain unknown.

If horizontal cortical projections for the agonist and antagonist muscles are topographically present in M1 in healthy people and PAS induces LTP- and LTD-like plasticities in M1 circuits, PAS may simultaneously change the cortical circuits controlling the agonist and antagonist muscles in M1. Specifically, we hypothesized that if the administration of PAS-25 ms with peripheral median nerve stimulation and TMS over agonist flexor carpi radialis (FCR) muscle representation topographically induces a simultaneous increase in cortical excitability for the agonist FCR muscle and a decrease in cortical excitability for the antagonist extensor carpi radialis (ECR) muscle in healthy people. We further hypothesized that GABA_B_-mediated cortical inhibitory functions for agonist FCR muscle may be specifically increased after PAS-25 ms in healthy people. Exploring how PAS affects the plasticity of M1 circuits for the agonist and antagonist muscles may increase our understanding of the organization processes of these muscles and expand previous findings. Therefore, this study aimed to investigate the changes in LTP- and LTD-like plasticity for agonist-antagonist muscle representations during PAS in healthy people.

## 2. Materials and Methods

### 2.1. Participants

To calculate the sample size, 80% statistical power, with an effect size of 0.35 and a two-sided α-level of 0.05 for the detection of changes in motor evoked potential (MEP) amplitudes, were used. G*Power 3.0 [38], based on these parameters, estimated a sample size of 13. Therefore, 15 healthy volunteers who were not taking any medication and without any risk of adverse events for TMS [39] were recruited (9 women, 6 men; age, 27.9 ± 10.6 [mean ± standard deviation (SD)] years; height, 163.6 ± 8.7 cm; weight, 60.1 ± 8.9 kg; and body mass index, 22.4 ± 2.6). The Edinburgh Handedness Inventory [40] confirmed right-handedness, with a mean laterality quotient of 1.0 ± 0.1 (mean ± SD) points.

Our experimental procedures were approved by the Research Ethics Committee of the Tokyo Kasei University (approval code SKE2019-7) and followed the principles of the Declaration of Helsinki. All participants provided written informed consent prior to participation. In addition, all experiments were performed following the “Guidelines for TMS/tES clinical services and research through the COVID-19 pandemic” [41].

### 2.2. Experimental Setup

Each participant was seated in a chair, with the right arm hanging to the side of the body trunk and the forearm held in place by a cushioned support with belts. The relaxed wrist could freely flex and extend. After wrist flexion and extension, the wrist gravitationally returned to the starting position. The left arm was comfortably placed on the armrest of the chair.

### 2.3. Electromyographic Recordings

The skin over the right flexor carpi radialis (FCR) muscle and extensor carpi radialis (ECR) muscle was cleaned with alcohol to reduce electrical resistance. The double differential surface electrodes (FAD-DEMG1, 4Assist, Tokyo, Japan) were then placed over the FCR and ECR muscles. The MEPs from the FCR and ECR muscles were amplified by 100, bandpass-filtered at 10–2000 Hz, digitized at 10 kHz with a PowerLab system (ADInstruments, Dunedin, New Zealand), and stored in a solid-state drive.

### 2.4. Transcranial Magnetic Stimulation

Participants wore a tight-fitting cap, and intersecting nasion–inion and interaural lines on the cap were marked with a pen to localize the vertex (Cz), in accordance with the international 10–20 system. The Magstim 200^2^ (Magstim, Whitland, UK) stimulators delivered TMS as a monophasic current waveform via a cable to the figure-of-eight coil on the participant’s head. To induce the current from the posterolateral to the anteromedial left brain, the coil was placed tangentially to the scalp, and the handle pointed backward and sideways at approximately 45° from the midline. The resting motor threshold (RMT) was determined as the minimum stimulus intensity required to produce a peak-to-peak MEP of at least 50 μV in each of the relaxed FCR or ECR muscles in 5 of the 10 consecutive trials.

### 2.5. Motor Representational Map

We did not use the magnetic resonance imaging (MRI) scanner and neuronavigation system because, although these systems can anatomically detect the motor hand area of the precentral gyrus [42,43], they could not anatomically distinguish agonist FCR and antagonist ECR muscle representations. Therefore, to map the agonist and antagonist muscle representations, 25 positions of a 6 × 6 cm^2^ grid with 1.5-cm spacings were marked on a participant’s head, with a center 6 mm anterior and 57 mm lateral from the Cz, based on previous studies [25]. At each scalp position, the MEPs evoked by five stimulations at 120% of the RMT of each FCR or ECR muscle were recorded in a clockwise spiral course of a 5-s interstimulus interval. We computed the center of amplitude (CA) of each FCR or ECR muscle to determine the coordinates with reference to Cz [17,24,25,26,44], according to Equation (1):(1)$$CA=\left[\begin{array}{cc} \frac{\sum a_{t}x_{t}}{\sum a_{t}} & \frac{\sum a_{t}y_{t}}{\sum a_{t}} \end{array}\right]$$
where t is the number of TMS, $x_{t}$ and $y_{t}$ are the TMS coordinates, and $a_{t}$ is the peak-to-peak MEP amplitude. CAs correspond to the locations of the relative excitable neuron populations projecting to each FCR or ECR muscle.

### 2.6. Paired Associative Stimulation

Prior to PAS, the threshold of perception for electrical stimulation was assessed. We placed the bipolar electrode of an electrical stimulator (DS3 Isolated Current Stimulator; Digitimer Ltd., London, UK) on the median nerve at the anterior part of wrist, with the cathode proximal to the anode. The threshold of perception was determined to be the minimum stimulus intensity that the participant first and once perceived with certainty as the stimulus, when electrical square-wave pulses with a 2000-μs duration were altered via 1-mA increments and decrements.

PAS was then delivered via the electric stimulation of 200 pairs of peripheral median nerves and TMS at the CA of the FCR muscle (frequency, 0.25 Hz) [6,17,45,46,47]. During PAS, square-wave pulses with a duration of 2000 μs at thrice the perceptual threshold were applied. The TMS applied to the left M1 at CA for the FCR muscle was evoked with 130% of the FCR muscle’s RMT. The interstimulus interval between the electrical stimulation and TMS was 25 ms (PAS-25 ms) and 10 ms (PAS-10 ms), as previous studies have noted that a 25-ms interstimulus interval was required to induce an LTP-like increment in corticospinal excitability [1,2,8,16,17,48], whereas a 10-ms interstimulus interval was required to induce a LTD-like reduction in cortical excitability [8,16,49,50,51,52]. PAS-25 ms and PAS-10 ms were performed on two different days, with a break of at least one day in between.

### 2.7. Cortical Excitability Recordings

Previous studies suggested that PAS may induce the reinforcement of cortical excitability in GABA_A_- and GABA_B_-ergic cortical circuits [9,17]. Therefore, unconditioned MEPs, GABA_A_-mediated short-interval intra-cortical inhibition (SICI), and GABA_B_-mediated long-interval intra-cortical inhibition (LICI) were measured at CA of each FCR (i.e., agonist) or ECR (i.e., antagonist) muscle before and after PAS-25 ms and PAS-10 ms [2,53,54,55]. Unconditioned MEPs were evoked at the CA of each FCR or ECR muscle at 120% of the RMT value. For SICI, the stimulus intensity for the first conditioning pulse was set at 80% of the FCR muscle’s RMT, whereas the second test pulse was set at 120% of the FCR muscle’s RMT. An interstimulus interval of 2.5 ms was used to test SICI [53,54,55]. For LICI, the stimulus intensity was set at 120% of the RMT for both the conditioning and test stimuli, and 100 ms was used as the interstimulus interval [2,56,57,58]. A total of 30 trials of unconditioned MEP, SICI, and LICI measurements (10 trials each) at a frequency of 0.2 Hz were recorded randomly.

### 2.8. Data Analysis

The outliers of the peak-to-peak MEP amplitudes were checked using Tukey’s fences, and values greater than 1.5 times the interquartile range of each participant’s initial dataset were excluded from the initial datasets [59]. The blank cells removed by the outliers were linearly interpolated. Then, the local linear trend model (LLT) was used to estimate the dynamic state of the peak-to-peak MEP amplitudes. The LLT assumes that the trend from observational values (Equation (2)), level (Equation (3)), and slope (Equation (4)) is calculated as follows:(2)$$o_{t}=l_{t}+\varepsilon_{t}$$
(3)$$l_{t+1}=l_{t}+s_{t}+u_{t}$$
(4)$$s_{t+1}=s_{t}+v_{t}$$
where t is the number of TMS, $o_{t}$ is the observational value, $\varepsilon_{t}$ is random variable, $l_{t}$ is the level, $s_{t}$ is the slope, and $u_{t}$ and $v_{t}$ are the disturbances in the level and slope, respectively [55].

#### 2.8.1. Immediate Modulatory Effects during PAS

Previous studies [60,61] noted that MEP amplitudes randomly fluctuated between TMS. We predicted that PAS would induce an increment in the MEP amplitudes with random fluctuation. Therefore, a linear regression model with a random fluctuation was constructed as follows:(5)$$y_{t}=\alpha +\beta_{t}+\varepsilon_{t}$$
where t is the number of PAS; α is the y-intercept of the peak-to-peak MEP amplitudes, reflecting initial cortical excitability; β is the slope of the peak-to-peak MEP amplitudes, reflecting changes in the cortical excitability; and $\varepsilon_{t}$ is the random variation, reflecting the inherent fluctuation of peak-to-peak MEPs.

We tested the fit of the time course of the mean actual peak-to-peak MEP amplitudes and used a linear regression model based on the coefficient of determination (R^2^). To evaluate the immediate or real-time effect of PAS-25 ms and PAS-10 ms, the slope (β) was extracted from the linear regression model.

#### 2.8.2. Retardative Modulatory Effects after PAS

To classify the change in cortical excitability after PAS as either retardative or delayed modulatory effects, the conditioned and unconditioned peak-to-peak MEP amplitudes (i.e., SICI and LICI) were normalized to the baseline data as follows:(6)$$NA\left(f,t\right)=\frac{D\left(f,t\right)-B\left(f\right)}{B\left(f\right)}$$
where NA denotes the normalized MEP amplitude, D denotes the actual peak-to-peak MEP amplitude at time t during PAS, and B denotes the mean actual peak-to-peak MEP amplitude before PAS. A large positive value indicates a large increase in the peak-to-peak MEPs compared with that in the baseline period before PAS.

### 2.9. Statistical Analysis

After normality testing using the Kolmogorov–Smirnov test, a parametric paired *t* test or a nonparametric Wilcoxon rank sum test was used to compare the slope of the cortical excitabilities for the FCR or ECR muscles induced by PAS-25 ms and PAS-10 ms for immediate modulatory effects during PAS and to compare the differences in normalized peak-to-peak MEP amplitudes among cortical excitabilities for the FCR or ECR muscles induced by PAS-25 ms and PAS-10 ms for retardative modulative effects after PAS. Data analysis was conducted using the SciPy package in the Python environment (Python Software Foundation, Wilmington, DE, USA), and R 3.4.0 (The R Foundation, Vienna, Austria). Data are expressed as the means ± standard errors of the mean (SEM). Statistical significance was set at *p* < 0.05.

## 3. Results

All participants completed all experiments without any side effects from PAS.

### 3.1. Motor Representational Map

The mean representational maps and CA values for the FCR and ECR muscles are shown in Figure 1. The maps for the FCR and ECR muscles overlapped and were changed after both PAS-25 ms and PAS-10 ms. The CA values for the FCR and ECR muscles are shown in Table 1. The CA values of the FCR and ECR muscles were not identical before and after both PAS-25 ms and PAS-10 ms.

### 3.2. Immediate Modulatory Effects during PAS

Figure 2 depicts the time-oriented changes in the unconditioned peak-to-peak MEP amplitudes of the FCR and ECR muscles as the immediate modulatory effects during PAS-25 ms and PAS-10 ms. Table 2 shows the α and β values in Equation (5). The linear regression model had moderate to high R^2^ values among the mean actual MEP amplitudes and predicted MEP amplitudes derived from the model (R^2^ = 0.150–0.722, all *p* < 0.0001). The β values of the regression model for ECR MEP during PAS-10 ms were significantly steeper than those during PAS-25 ms (Wilcoxon rank sum test, *p* = 0.002). Although the β values of the regression model for FCR MEP during PAS-10 ms were steeper than those during PAS-25 ms, the results were not significant (Wilcoxon rank sum test, *p* = 0.412).

### 3.3. Retardative Modulatory Effects after PAS

Peak-to-peak MEP amplitudes and values of RMT, SICI, and LICI obtained for the FCR and ECR muscles before and after PAS are shown in Table 3 and Figure 3. Group NA values for the FCR and ECR muscles before and after PAS-25 ms and PAS-10 ms are shown in Figure 4. The Kolmogorov–Smirnov test demonstrated that the MEP amplitude lacked normality (each NA value for the FCR or ECR muscle before and after PAS-25 ms or PAS-10 ms: all *p* < 0.05). Therefore, nonparametric testing was used for comparison of the NA values for each FCR or ECR muscles after PAS-25 ms and PAS-10 ms treatments. The Wilcoxon rank sum test showed the NA values of the unconditioned MEP for FCR for PAS-25 ms were significantly larger than those for PAS-10 ms, whereas the NA values of the unconditioned MEP for ECR for PAS-25 ms were significantly smaller than those for PAS-10 ms (Wilcoxon rank sum test: FCR, *p* = 0.001; ECR, *p* < 0.0001). The NA values of SICI for FCR for PAS-25 ms were significantly smaller than those for PAS-10 ms, and the NA values of SICI for ECR for PAS-25 ms were also significantly smaller than those for PAS-10 ms (Wilcoxon rank sum test: FCR, *p* = 0.002; ECR, *p* < 0.0001). The NA values of LICI for ECR for PAS-25 ms were significantly smaller than those for PAS-10 ms (Wilcoxon rank sum test: *p* < 0.0001). Although the NA values of LICI for FCR were also smaller than those for PAS-10 ms, significance was not reached (Wilcoxon rank sum test: FCR, *p* = 0.451).

## 4. Discussion

We measured the changes in the MEP amplitude related to the M1 circuits’ plasticity for the agonist and antagonist muscles, induced by PAS-25 ms and PAS-10 ms to test the hypothesis that PAS should topographically induce an increment of cortical excitability for the agonist FCR muscle and a decrement of cortical excitability for the antagonist ECR muscle in healthy people. This method was also used to test whether GABA_B_-mediated cortical inhibitory functions for the agonist FCR muscle were specifically increased after PAS-25 ms in healthy people. The results of this study revealed the following: (a) the MEP amplitudes of both the agonist (FCR) and antagonist (ECR) muscles were increased during PAS-10 ms, (b) the unconditioned MEP amplitudes for the agonist FCR muscle after PAS-25 ms were larger than those after PAS-10 ms, whereas the unconditioned MEP amplitudes for the antagonist ECR muscle after PAS-25 ms were smaller than those after PAS-10 ms, and (c) the GABA_A_-mediated SICI and GABA_B_-mediated LICI for both the agonist and antagonist muscles after PAS-25 ms were smaller than those after PAS-10 ms. These systematic observations provided evidence that PAS-25 ms and PAS-10 ms had different effects on the LTP- and LTD-like plasticity for the agonist and antagonist muscles.

The first additional new observation in our study was that the cortical excitabilities for the agonist and antagonist muscles increased more during PAS-10 ms than PAS-25 ms as an immediate effect. Several studies [2,8,16] have suggested that cortical excitability for the agonist muscle increased and decreased during PAS-25 ms and PAS-10 ms, respectively. However, some studies have reported no such changes in excitability during PAS-25 ms and PAS-10 ms treatment [46,62,63], while another noted contradictory effects of PAS: a decrement of cortical excitability during PAS-25 ms and an increment of cortical excitability during PAS-10 ms [64]. Therefore, the effects of PAS-25 ms and PAS-10 ms remain a topic of contention. One of the reasons for these inconsistencies may be the inherent random fluctuation of MEPs [60,61] and changing the representational maps with PAS. The coil position for appropriately eliciting MEPs from the agonist and antagonist muscles was systematically determined using CA. Relating to the systematic MEP measurements, to induce the topographical modulatory effects of PAS, PAS-25 ms and PAS-10 ms were composed by the pairing of peripheral median nerve stimulation and TMS at the CA for the agonist FCR muscle. As a result, PAS was applied with the appropriate TMS coil position. Additionally, to detect the dynamic states of cortical excitability, we used the LLT model, thus eliminating the confounding factor of amplitude fluctuations during MEP evaluations. These were thought to be the basis for successfully observing cortical excitability for the agonist and antagonist muscles during PAS. In fact, our study noted that the MEP amplitudes of the agonist (FCR) and antagonist (ECR) muscles were increased more during PAS-10 ms than PAS-25 ms. Homeostatic plasticity, based on the Bienenstock–Cooper–Munro theory [1,65,66], assumes a bidirectional synaptic plasticity, with the threshold for the induction of LTP and LTD varying. According to the Bienenstock–Cooper–Munro theory, the threshold decreases with low levels of previous postsynaptic activity, indicating LTP induction, and vice versa with high levels of previous postsynaptic activity, indicating LTD induction. One possible explanation for the increase and decrease in cortical excitabilities during PAS-10 ms and PAS-25 ms, respectively, in our study, is that the threshold decreases with low levels of postsynaptic activity during PAS-10 ms, but the threshold increases with high levels of postsynaptic activity during PAS-25 ms.

In retardative modulatory effects, previous studies [2,6] reported that unconditioned MEP amplitudes for the targeted agonist muscle increased more than those for the non-target muscle after PAS, indicating the topographic specificity of PAS. Furthermore, a previous study [10] reported that MEP amplitudes for the targeted agonist muscles increased, whereas those for non-targeted antagonist muscles were not significantly changed before and after PAS. In expanding on the findings of previous studies [6,10], the unconditioned MEP amplitudes for the agonist (FCR) and antagonist (ECR) muscles were reciprocally more and less increased after PAS-25 ms, whereas they were reciprocally less and more increased after PAS-10 ms as a retardative effect in our study. This is the second new finding in our study. A previous study [22] reported horizontal axon collaterals connecting different forelimb representations in M1. The representational map areas for the agonist and antagonist muscles overlap [23,24]. These studies suggest that the horizontal cortical projections interconnecting functionally-related neuronal clusters within M1 regulate the agonist and antagonist muscles, and that the somatosensory cortex inputs to M1 may diverge onto the FCR and ECR muscle representations due to peripheral electrical stimulation [22,23,24]. As a result, M1 facilitates the passage of Ia inhibitory interneurons from the corticospinal tract or inhibitory volleys that travel from M1 to the antagonist muscle motor neurons [24,29,30]. One possible explanation for reciprocally increased and decreased MEP amplitudes for the agonist FCR and antagonist ECR muscles in our study is that peripheral median nerve electrical stimulation provides an input to the FCR representation in M1 via afferents from the somatosensory cortex. Consequently, the output from the FCR representation might diverge onto alpha motor neurons in FCR muscles and Ia inhibitory interneurons in ECR muscles. Then, PAS influences reciprocal inhibition functions via LTP- and LTD-like plasticity of the M1 circuit. Furthermore, the retardative and immediate modulatory effects of PAS on unconditioned MEPs may be a different mechanism of transient homeostatic plasticity and a continuous reciprocal inhibitory function.

In our study, PAS-10 ms decreased (i.e., disinhibited) the GABA_A_- and GABA_B_-mediated cortical inhibitory function (i.e., LICI) for both the agonist (FCR) and antagonist (ECR) muscles, whereas PAS-25 ms decreased it less. Several previous studies have investigated the aftereffects of a PAS intervention on both the GABA_A_- and GABA_B_-ergic cortical circuits [2,13,67]. Their results suggested that PAS may induce the selective reinforcement of GABA_B_-ergic cortical circuits. A previous study [14] reported that LICI was reduced by administering PAS-25 ms and increased by PAS-10 ms. Other study [68] found no significant changes in LICI arising from the application of a PAS-10 ms intervention. Additionally, a previous study [69] reported that experimentally induced LTP at cortical synapses can be reversed by blocking the GABA_A_ receptors. Although LTP occurs at excitatory glutamatergic synapses, GABA plays a significant role in modulating LTP [16] in the context of PAS. One possible explanation for the decrease in SICI and LICI after PAS-10 ms for both FCR and ECR is that GABA_A_- and GABA_B_-mediated cortical inhibitory functions for both agonist and antagonist muscles were less topographically decreased. However, the precise mechanism underlying the differences in the changes in the GABA_A_- and GABA_B_-ergic cortical plasticity for agonist and antagonist muscles induced by PAS-25 ms and PAS-10 ms is still unclear. Therefore, further studies are needed to investigate the precise role of GABA_A_-and GABA_B_-ergic M1 plasticity in topographic specificity during PAS-25 ms and PAS-10 ms.

This study has some limitations that should be acknowledged. First, CA is not the location inducing the peak MEP amplitude reflecting the most excitable neuron populations projecting to each FCR or ECR muscle, but rather the equilibrating location where the MEP amplitude is evenly distributed, reflecting relative excitable neuron populations that are projecting to each FCR or ECR muscle [17,24,26,44]. Therefore, further research is needed to investigate the precise location of the agonist and antagonist muscle representations, using not only TMS but also the neuronavigation system, MRI, and magnetoencephalography [70]. Second, the intensity of electrical stimulation and TMS is set to three times the perceptual threshold and 130% of the RMT, respectively; the interval between the peripheral nerve stimulus and the TMS pulse is set to 25 ms for inducing LTP and 10 ms for inducing LTD, based on previous studies [2,8,16]. However, a previous study found that PAS has highly individual-dependent effects [7,63]. In our study, although the sample size was estimated using G*Power 3.0 [38], we did not consider individual factors, such as adequate stimulus intensities and the interstimulus interval of the electrical stimulation and TMS in accordance with the differences in age, sex, and height. Thus, future studies need to include a larger sample size to analyze the effects of PAS while taking individual factors into account.

## 5. Conclusions

In conclusion, despite some potential limitations in our study for the non-homogeneity between age, sex, and height, due to a small sample size, we found LTP- and LTD-like M1 plasticity for agonist and antagonist muscles via PAS-25 ms and PAS-10 ms. Our findings suggest that during PAS-10 ms, cortical excitability for both the agonist and antagonist muscles was less topographically increased during PAS, implying an immediate homeostatic effect. However, the unconditioned MEP amplitudes for the agonist FCR and antagonist ECR muscles were reciprocally more and less increased after PAS-25 ms, whereas they were reciprocally less and more increased after PAS-10 ms, implying retardative effects on the reciprocal inhibition. Additionally, after PAS-10 ms, the GABA_A_- and GABA_B_-mediated cortical inhibitory functions for both the agonist and antagonist muscles were less topographically decreased, implying a retardative disinhibition effect.

## Figures and Tables

**Figure 1 brainsci-13-00475-f001:**
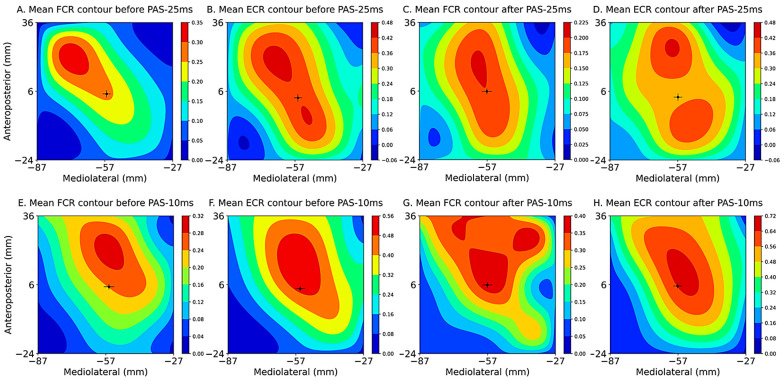
Mean contours of the FCR (**A**) and ECR (**B**) muscles before PAS-25 ms, those of FCR (**C**) and ECR (**D**) muscles after PAS-25 ms, those of the FCR (**E**) and ECR (**F**) muscles before PAS-10 ms, and those of the FCR (**G**) and ECR (**H**) muscles after PAS-10 ms. The color code of each contour denotes the MEP amplitudes. Black circles denote the mean CA, and the error bar denotes the SEM. The vertex (Cz) reflects the coordinate origin. FCR, flexor carpi radialis; ECR, extensor carpi radialis; PAS-25 ms, paired associative stimulation with a 25-ms interstimulus interval between peripheral electrical stimulation and TMS; PAS-10 ms, paired associative stimulation with a 10-ms interstimulus interval between peripheral electrical stimulation and TMS; MEP, motor evoked potential; SEM, standard error of the mean.

**Figure 2 brainsci-13-00475-f002:**
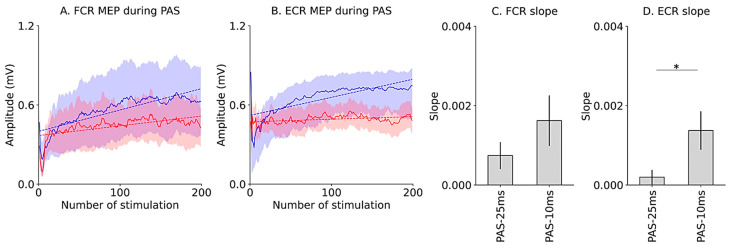
Grand-averaged time-series of the MEP amplitudes of FCR (**A**) and ECR (**B**) during PAS. The red and blue solid lines indicate the mean actual MEP amplitudes during PAS-25 ms and PAS-10 ms, respectively. The colored areas indicate the standard error of the mean. The red and blue dashed lines indicate the mean predicted MEP amplitudes during PAS-25 ms and PAS-10 ms by the linear regression model, respectively. The slopes of the MEP amplitudes for the FCR (**C**) and ECR (**D**) muscles during PAS. The slopes of MEP amplitudes for the ECR muscle were significantly steeper during PAS-10 ms than those during PAS-25 ms. MEP, motor evoked potential; FCR, flexor carpi radialis; ECR, extensor carpi radialis; PAS-25 ms, paired associative stimulation with a 25-ms interstimulus interval between peripheral electrical stimulation and TMS; PAS-10 ms, paired associative stimulation with a 10-ms interstimulus interval between peripheral electrical stimulation and TMS. *: *p* < 0.05.

**Figure 3 brainsci-13-00475-f003:**
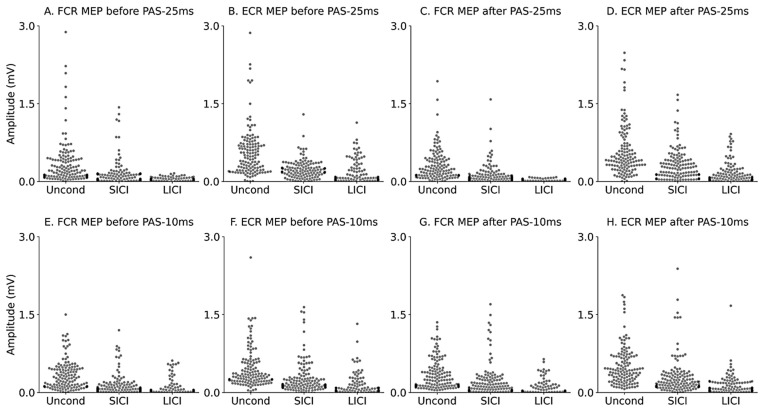
Scatter charts of MEP amplitudes for FCR (**A**) and ECR (**B**) before PAS-25 ms, those for FCR (**C**) and ECR (**D**) after PAS-25 ms, those for FCR (**E**) and ECR (**F**) before PAS-10 ms, and those for FCR (**G**) and ECR (**H**) after PAS-10 ms at the individual level. Unconditioned MEP amplitudes were higher than the conditioned MEP amplitudes (i.e., SICI and LICI conditions). MEP, motor evoked potential; FCR, flexor carpi radialis; ECR, extensor carpi radialis; PAS-25 ms, paired associative stimulation with a 25-ms interstimulus interval between peripheral electrical stimulation and TMS; PAS-10 ms, paired associative stimulation with a 10-ms interstimulus interval between peripheral electrical stimulation and TMS; Uncond, unconditioned MEP amplitude; SICI, short-interval intracortical inhibition; LICI, long-interval intracortical inhibition.

**Figure 4 brainsci-13-00475-f004:**
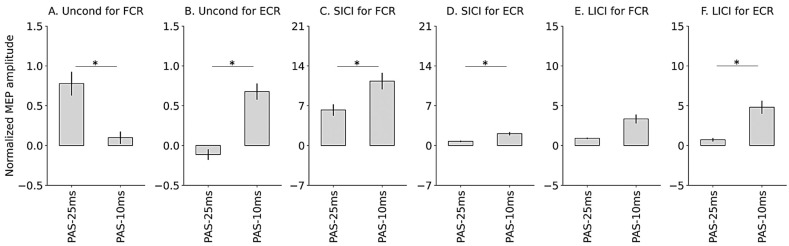
Bar graphs of NA values for the unconditioned MEP for FCR (**A**) and ECR (**B**) muscles, SICI for FCR (**C**) and ECR (**D**) muscles, and LICI for FCR (**E**) and ECR (**F**) after PAS-25 ms and PAS-10 ms at the group level. The unconditioned MEP for FCR for PAS-25 ms were larger than those for PAS-10 ms, whereas the unconditioned MEP for ECR for PAS-25 ms were smaller than those for PAS-10 ms. The SICI and LICI for both FCR and ECT for PAS-25 ms were smaller than those for PAS-10 ms. NA, the normalized motor evoked potential amplitude; MEP, unconditioned motor evoked potential; Uncond, unconditioned MEP amplitude; SICI, short-interval intracortical inhibition; LICI, long-interval intracortical inhibition. *: *p* < 0.05.

**Table 1 brainsci-13-00475-t001:** Center of amplitude of peak-to-peak MEP amplitudes before and after PAS.

	Center of Amplitude (mm)			
	FCR		ECR	
	Anteroposterior	Mediolateral	Anteroposterior	Mediolateral
Before PAS-25 ms Before PAS-10 ms After PAS-25 ms After PAS-10 ms	−56.2 ± 1.6 −55.6 ± 1.2 −57.8 ± 1.4 −57.1 ± 1.2	4.9 ± 1.7 5.1 ± 2.3 4.3 ± 1.8 5.9 ± 2.2	−55.7 ± 1.6 −55.0 ± 1.1 −56.9 ± 1.6 −57.2 ± 1.0	3.1 ± 1.8 4.3 ± 2.2 3.5 ± 1.9 5.4 ± 2.0

The values are the mean ± standard error of the mean. The vertex (Cz) reflects the coordinate origin. MEP, motor evoked potential; PAS-25 ms, paired associative stimulation with a 25-ms interstimulus interval between peripheral electrical stimulation and TMS; PAS-10 ms, paired associative stimulation with a 10-ms interstimulus interval between peripheral electrical stimulation and TMS; FCR, flexor carpi radialis; ECR, extensor carpi radialis.

**Table 2 brainsci-13-00475-t002:** Assessment of the model fit.

Condition	Muscle	α	β	R^2^	*p*
PAS-25 ms	FCR ECR	0.367 ± 0.138 0.472 ± 0.088	0.001 ± 0.000 0.000 ± 0.000	0.411 0.150	<0.0001 <0.0001
PAS-10 ms	FCR ECR	0.399 ± 0.197 0.520 ± 0.111	0.002 ± 0.001 0.001 ± 0.000	0.722 0.644	<0.0001 <0.0001

The values are the mean ± standard error of the mean.

**Table 3 brainsci-13-00475-t003:** MEP amplitudes and RMT, SICI, and LICI values for the FCR and ECR muscles before and after PAS.

	Before PAS-25 ms		After PAS-25 ms		Before PAS-10 ms		After PAS-10 ms	
	FCR	ECR	FCR	ECR	FCR	ECR	FCR	ECR
RMT (%)	58.3 ± 2.5	56.6 ± 1.9	57.0 ± 1.9	54.6 ± 1.6	58.7 ± 2.3	55.3 ± 1.8	59.6 ± 2.1	56.1 ± 1.7
Uncond (mV)	0.33 ± 0.03	0.53 ± 0.04	0.31 ± 0.03	0.54 ± 0.04	0.34 ± 0.02	0.43 ± 0.03	0.34 ± 0.02	0.47 ± 0.03
SICI (mV)	0.13 ± 0.02	0.22 ± 0.02	0.14 ± 0.02	0.29 ± 0.02	0.15 ± 0.02	0.26 ± 0.02	0.20 ± 0.02	0.31 ± 0.03
LICI (mV)	0.04 ± 0.00	0.15 ± 0.02	0.03 ± 0.00	0.17 ± 0.02	0.08 ± 0.01	0.14 ± 0.02	0.08 ± 0.01	0.14 ± 0.02

The values are the mean ± standard error of the mean. MEP, motor evoked potential; RMT, resting motor threshold; SICI, short-interval intra-cortical inhibition; LICI, long-interval intra-cortical inhibition; FCR, flexor carpi radialis; ECR, extensor carpi radialis; PAS, paired associative stimulation; Uncond, unconditioned MEP amplitudes.

## Data Availability

Raw data were generated at Tokyo Kasei University. Derived data supporting the findings of this study are available from the corresponding author, M.S.

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
