# Peer review of "Effects of Paired Associative Stimulation on Cortical Plasticity in Agonist–Antagonist Muscle Representations"

_brainsci, 2023, doi:10.3390/brainsci13030475_

Round 1

Reviewer 1 Report

Online and offline cortical plasticity for reciprocal muscle representation by PAS

The authors studied 200 pairs of 0.25 Hz of PAS (peripheral electric stimulation of the right median nerve at the wrist followed by transcranial magnetic stimulation of the left M1 with ISI interval of 25 ms (PAS-LTP) and 10 ms (PAS-LTD). Their results indicate that cortical excitability for the agonist and antagonist muscles was reciprocally and topographically increased and decreased after PAS, but GABAB-mediated cortical inhibitory functions for both the agonist and antagonist muscles were less topographically decreased after PAS-LTD. 

-The title is confusing:

1) Online and offline  cortical plasticity? Difficult to understand 

2) Why did the authors use the terminology "reciprocal muscle representation"? What is the definition of this terminology? Why not agonist-antagonist muscle?

Abstract:

PAS-LTP and PAS-LTD is not correctly used: it should be used "PAS-25ms and PAS-10ms" because the authors studied PAS effect over agonist-antagonist muscle and they did not know which muscle was potentiated, which was depressed.

-The result: nothing is clear: there is not any number, neither any comparison, neither p value. 

-L. 24: "...more or less increased...", which result is that? Significantly increased? In which muscle? 

Introduction

-Hypothesis should be more clear and it should be emphasize that it will be done in healthy subject".

Expert should review statistical analysis

Methods:

It should be clearer: which variable they measured

L. 130, how many recording was done to determine the perception threshold? Once?

When were the LTP and LTD realised? the same day? 

Data analysis

-How was the amplitude measured? Peak to peak? 

Result:

-One table with all rough data could be very informative: Threshold of ECR and  FCR, MEP amplitude, SICI, LICI at baseline and after intervention etc.

-The graphics should be reorganized better as figure 2: FCR before and after 25ms and 10ms could be put in one graphic and the same for ECR.

-Figure 4: it is not clear why the authors compare FCR data with ECR? such as ECR-PAS-LTP with FCR-PAS-LTD? 

-In the figures legends many number could go to tables.

Discussion

The authors try to explain their result with cortical TMS over hot point of FCR. How about peripheral nerve topography? Could it have a different effect over FCR-ECR? 

Author Response

Manuscript ID: brainsci-2253441
Title: Online and offline cortical plasticity for reciprocal muscle representations by paired
associative stimulation

Point-by-Point Response

Thank you very much for your favorable comments on our study and your helpful suggestions.
Our responses to your comments are as presented below. Please note that the changes made do
not influence the content, conclusions, or framework of the paper. We have not listed below all
minor changes made; however, these are underlined in the revised manuscript.

Responses to Reviewer 1

Comment 1

Online and offline cortical plasticity? Difficult to understand.

Response to comment 1

As you noted, the terms “online” and “offline” are unclear. We have therefore rewritten the title:

Cortical plasticity for agonist-antagonist muscle representations by paired associative
stimulation

We have also changed the terms of “online” and “offline” into “the immediate or real-time” and
retardative or delayed” throughout the manuscript, respectively.

Comment 2

Why did the authors use the terminology "reciprocal muscle representation"? What is the
definition of this terminology? Why not agonist-antagonist muscle?

Response to comment 2

As you noted, the term of “reciprocal muscle” is unclear. We have therefore changed the term
from “reciprocal muscle” to “agonist and antagonist muscle” throughout the manuscript.

Comment 3

Abstract: PAS-LTP and PAS-LTD is not correctly used: it should be used "PAS-25ms and
PAS-10ms" because the authors studied PAS effect over agonist-antagonist muscle and they
did not know which muscle was potentiated, which was depressed.

2
Response to comment 3

As you noted, the terms of “PAS-LTP and PAS-LTD are not correct. We have therefore
changed the term of “PAS-LTP and PAS-LTD” into “PAS-25ms” and PAS-10ms”
throughout the manuscript.

Comment 4

-The result: nothing is clear: there is not any number, neither any comparison, neither p
value.

-L. 24: "...more or less increased...", which result is that? Significantly increased? In which
muscle?

Response to comment 4

We appreciate your helpful comment and have revised the sentences in the Abstract (page 1,
lines 2328) as follows:

The unconditioned motor evoked potential (MEP) amplitudes of agonist muscles for PAS-25ms
were significantly larger than those for PAS-10ms (p = 0.001), whereas the unconditioned MEP
amplitudes of antagonist muscles for PAS-25ms were significantly smaller than those for PAS-10ms
(p < 0.0001). The γ-aminobutyric acid A (GABAA)- and GABAB-mediated cortical inhibitions for
agonist and antagonist muscles for PAS-25ms were increased more than those for PAS-10ms.

Comment 5

Introduction: Hypothesis should be more clear and it should be emphasize that it will be done
in healthy subject".

Response to comment 5

As you noted, it is important to clearly explain the hypothesis and to emphasize the heathy
subject in our study. We have therefore revised the sentences in the Introduction (page 2, lines
4687) as follows:

Specifically, if the peripheral input precedes M1 stimulation due to peripheral electrical
nerve stimulation at 25 ms (PAS-25ms) before transcranial magnetic stimulation (TMS) over
M1, PAS can result in an increased cortical excitability (LTP-like plasticity) [1,8,9]. In contrast,
if the peripheral input follows M1 stimulation due to peripheral electrical nerve stimulation
at 10 ms (PAS-10ms) before TMS over M1, cortical excitability is reduced (long-term
depression (LTD)-like plasticity) [1,2]. Additionally, previous studies [2,6,10] suggested that
the synaptic plasticity induced by PAS follows topographical rules because the homologous
connection of the somatosensory cortex and the M1 has high topographical specificity [11].
Furthermore, previous studies [12-15] also suggested that PAS induces selective
reinforcement of γ-aminobutyric acid B (GABAB)-ergic cortical circuits. These topography
specific LTP-like and LTD-like plasticity of PAS persists after stimulation [9,16,17]. Therefore,

3
PAS is applicable to patients suffering from motor control deficits, such as stroke [18],
Parkinson’s disease [19,20], and Huntington’s disease [21].

In motor control, layers III and V of M1 bidirectionally connect different muscle
representations [22]. Previous studies noted that the output from the common M1 site could
diverge onto agonist and antagonist muscles with different “gain” [17,23-26], and
stimulation of M1 region could elicit particular movements [27,28]. Therefore, movements
including agonist and antagonist muscles could be controlled by a network within the M1
[22,23]. Specifically, control of agonist-antagonist muscles is crucial for smooth and precise
motor control [24,29,30]. Several studies have suggested that the motor representations of
M1 can be reorganized during motor training [31-35], and that the competition with the
antagonist muscle representations would occur by the expansion of the trained agonist
muscle representation [31,36,37]. However, the changes in representations for agonist and
antagonist muscles after PAS remain unclear. Therefore, despite the fact that functional
interaction in M1 is crucial for human movement, the changes induced by PAS in cortical
circuits for agonist-antagonist muscle representations of M1 remain unknown.

If horizontal cortical projections for agonist and antagonist muscles are topographically
present in M1 in healthy people, and PAS induces LTP- and LTD-like plasticities in M1
circuits, PAS may be able to simultaneously change the cortical circuits controlling agonist
and antagonist muscles in M1. Specifically, we hypothesized that (1) if the PAS-25ms with
peripheral median nerve stimulation and TMS over agonist flexor carpi radialis (FCR)
muscle representation, then PAS should topographically induce an increment of cortical
excitability for agonist FCR muscle and simultaneously induce a decrement of cortical
excitability for antagonist extensor carpi radialis (ECR) muscle in healthy people. We further
hypothesized that (2) GABAB-mediated cortical inhibitory functions for agonist FCR muscle
was specifically increased after PAS-25ms in healthy people. Exploring how PAS affects the
M1 circuits plasticity for agonist and antagonist muscles may contribute to understanding
the organization processes for agonist and antagonist muscles by PAS, in addition to
expanding on previous findings. Therefore, we investigated changes in LTP- and LTD-like
plasticity for agonist-antagonist muscle representations during PAS in healthy people.

We have also added the following references:

36. Siebner, H.R.; Rothwell, J. Transcranial magnetic stimulation: new insights into representational
cortical plasticity. Exp Brain Res 2003, 148, 1-16.

37. Meesen, R.L.; Cuypers, K.; Rothwell, J.C.; Swinnen, S.P.; Levin, O. The effect of long-term TENS
on persistent neuroplastic changes in the human cerebral cortex. Hum Brain Mapp 2011, 32,
872-882.

Comment 6

Methods: It should be clearer: which variable they measured.

L. 130, how many recording was done to determine the perception threshold? Once?

4
Response to comment 6

As you noted, we should clearly explain how to determine the perception threshold. We have
therefore revised the sentences in the Paired Associative Stimulation” subsection of the
Materials and Methods (page 4, lines 146150) as follows:

We placed the bipolar electrode over the median nerve at the anterior part of wrist with the
cathode proximal. The threshold of perception of perception was determined to be the minimum
stimulus intensity required that the participant first perceived with certainty that the stimulus,
when the electrical square-wave pulses of 2000-μs duration, was altered in 1 mA increments and
decrements.

Comment 7

When were the LTP and LTD realised? the same day?

Response to comment 7

As you noted, we should clearly explain the schedule of PAS. We have therefore revised the
sentence in the “Paired Associative Stimulation” subsection of the Materials and Methods (page
4, lines 157158) as follows:

“PAS-25ms and PAS-10ms were performed on two different days with a break of at least one day
in between.”

Comment 8

Data analysis: How was the amplitude measured? Peak to peak?

Response to comment 8

As you noted, we measured peak-to-peak MEP amplitudes. We have therefore revised the
sentences in the “Data analysis” subsection of the Materials and Methods (page 4, lines 174
176; lines 193196; lines 205207; and lines 209215) as follows:

Outliers of peak-to-peak MEP amplitudes were checked by Tukey’s fences, and values greater
than 1.5 times interquartile range of each participant’s initial dataset were excluded from the initial
datasets [59].

where ? is the number of PAS, ? is the y-intercept of the peak-to-peak MEP amplitudes,
reflecting the initial cortical excitability; ? is the slope of the peak-to-peak MEP amplitudes,
reflecting changes in the cortical excitability; and ?? is the random variation reflecting the inherent
fluctuation of peak-to-peak MEPs.

For evaluating change in cortical excitability after PAS as retardative or delayed modulatory
effects, conditioned and unconditioned peak-to-peak MEP amplitudes (i.e., SICI and LICI) were

5
normalized to the baseline data (Equation 6).

where ?? denotes the normalized MEP amplitude, ? denotes actual peak-to-peak MEP
amplitude at time ? during PAS, and ? denotes the mean actual peak-to-peak MEP amplitude
before PAS. A large positive value indicates a large increase in peak-to-peak MEPs compared with
that in the baseline period before PAS. After normality testing using the KolmogorovSmirnov test,
either a parametric paired t test or a nonparametric Wilcoxon rank sum test was used to compare
differences in normalized peak-to-peak MEP amplitudes among cortical excitabilities for FCR or
ECR muscles by PAS-25ms and PAS-10ms.

Comment 9

Result: One table with all rough data could be very informative: Threshold of ECR and
FCR, MEP amplitude, SICI, LICI at baseline and after intervention etc.

Response to comment 9

We appreciate your helpful comment and added the Table 3 into the “Retardative modulatory
effects after PAS” subsection of the Results (page 9).

Comment 10

-The graphics should be reorganized better as figure 2: FCR before and after 25ms and 10ms
could be put in one graphic and the same for ECR.

Response to comment 10

We appreciate your helpful comment. We revised the Figure 2 in the Immediate modulatory
effects during PAS” subsection of the Results (page 9).

Comment 11

-Figure 4: it is not clear why the authors compare FCR data with ECR? such as ECR-PAS-
LTP with FCR-PAS-LTD?

Response to comment 11

As you noted, we should separately compare the peak-to-peak MEP amplitudes in each of either
FCR or ECR muscle. We have therefore revised the sentences in the “Data analysis” subsection
of the Materials and Methods (page 5, lines 200203; and lines 212215) as follows:

After normality testing using KolmogorovSmirnov testing, either a parametric paired t test or a
nonparametric Wilcoxon rank sum test was used to compare the slope among cortical excitabilities
for FCR or ECR muscles by PAS-25ms and PAS-10ms.

After normality testing using the KolmogorovSmirnov test, either a parametric paired t test or a

6
nonparametric Wilcoxon rank sum test was used to compare differences in normalized peak-to-
peak MEP amplitudes among cortical excitabilities for FCR or ECR muscles by PAS-25ms and PAS-
10ms.

We revised the sentences in Immediate modulatory effects during PAS” subsection of the
Results (page 7, lines 260264) as follows:

The ? values of regression model for ECR MEP during PAS-10ms were significantly steeper than
those during PAS-25ms (Wilcoxon rank sum test, p = 0.002). Although the ? values of regression
model for FCR MEP during PAS-10ms were steeper than those during PAS-25ms, the results were
not significant (Wilcoxon rank sum test, p = 0.412).

We revised the Retardative modulatory effects after PAS” subsection of the Results (page 8,
lines 290301) as follows:

Therefore, nonparametric testing was used for comparison of the ?? values for each FCR or ECR
muscles after PAS-25ms and PAS-10ms treatments. The Wilcoxon rank sum test showed the ??
values of unconditioned MEP for FCR for PAS-25ms were significantly larger than those for PAS-
10ms, whereas the ?? values of unconditioned MEP for ECR for PAS-25ms were significantly
smaller than those for PAS-10ms (Wilcoxon rank sum test: FCR, p = 0.001; ECR, p < 0.0001). The
?? values of SICI for FCR for PAS-25ms were significantly smaller than those for PAS-10ms, and
the ?? values of SICI for ECR for PAS-25ms were also significantly smaller than those for PAS-
10ms (Wilcoxon rank sum test: FCR, p = 0.002; ECR, p < 0.0001). The ?? values of LICI for ECR
for PAS-25ms were significantly smaller than those for PAS-10ms (Wilcoxon rank sum test: p <
0.0001). Although the ?? values of LICI for FCR were also smaller than those for PAS-10ms,
significance was not reached (Wilcoxon rank sum test: FCR, p = 0.451).

And, we revised the Figure 4 in the Retardative modulatory effects after PAS” subsection of
the Results (page 10).

Comment 12

Discussion: The authors try to explain their result with cortical TMS over hot point of FCR.
How about peripheral nerve topography? Could it have a different effect over FCR-ECR?

Response to comment 12

As you noted, it is important to explain how the peripheral nerve topography affects the effect
of PAS on FCR and ECR MEP amplitude. We have therefore revised the sentences in the
Introduction (page 2, lines 5159; and lines 7387) as follows:

Additionally, previous studies [2,6,10] suggested that the synaptic plasticity induced by PAS
follows topographical rules because the homologous connection of the somatosensory cortex and
the M1 has high topographical specificity [11]. Furthermore, previous studies [12-15] also

7
suggested that PAS induces selective reinforcement of γ-aminobutyric acid B (GABAB)-ergic
cortical circuits. The topography specific LTP-like and LTD-like plasticity of PAS persists after
stimulation [9,16,17]. Therefore, PAS is applicable to patients suffering from motor control deficits,
such as stroke [18], Parkinson’s disease [19,20], and Huntington’s disease [21].

If horizontal cortical projections for agonist and antagonist muscles are topographically present in
M1 in healthy people, and PAS induces LTP- and LTD-like plasticities in M1 circuits, PAS may be
able to simultaneously change the cortical circuits controlling agonist and antagonist muscles in
M1. Specifically, we hypothesized that (1) if the PAS-25ms with peripheral median nerve
stimulation and TMS over agonist flexor carpi radialis (FCR) muscle representation, then PAS
should topographically induce an increment of cortical excitability for agonist FCR muscle and

simultaneously induce a decrement of cortical excitability for antagonist extensor carpi radialis
(ECR) muscle in healthy people, and that (2) GABAB-mediated cortical inhibitory functions for
agonist FCR muscle was specifically increased after PAS-25ms in healthy people.

We revised the sentences in the Discussion (page 10, line 360page 11, line 363; page 11, lines
394401) as follows:

Relating to systematic MEP measurements, to induce topographical modulatory effects of PAS,
PAS-25ms and PAS-10ms were composed by pairing of peripheral median nerve stimulation and
TMS at the ?? for the agonist FCR muscle. As a result, PAS was applied with the appropriate TMS
coil position.

One possible explanation for reciprocally increased and decreased MEP amplitudes for agonist
FCR and antagonist ECR muscles in our study is that peripheral median nerve electrical
stimulation provides an input to the FCR representation in M1 via afferents from the
somatosensory cortex. Consequently, the output from the FCR representation might diverge and
facilitate onto alpha motor neurons to FCR and Ia inhibitory interneurons to ECR muscles. Then,
PAS influences reciprocal inhibition functions via LTP- and LTD-like plasticity of the M1 circui

Reviewer 2 Report

The manuscript turns out to be very interesting and deals with a very topical topic. However, I have some concenrs.

I think the introduction paragraph needs to be rewritten as it is too short and the purpose of the study is not clear.

The anthropometric characteristics of the subjects recruited should be better explained in the methods. As they are written they are difficult to understand.

Furthermore, still in the methods, the TMS procedure should be better explained. In particular, the positioning and the tools used should be well explained (Headset eeg?? Neuro navigator with scalp reconstruction?).

In addition, the physiological mechanisms should be better explained in the discussions, and finally the limits of the study should be reported in the "conclusions" paragraph (non-homogeneity of the sample between males/females; low number, etc...)

Reviewer 3 Report

This study uses paired associative stimulation of the hand area of the motor cortex to induce LTP and LTD changes in reciprocal muscles, specifically FCR and ECR, to explore the organisation process for reciprocal muscles.

The Introduction is comprehensive and well written

The Methods section is clearly written and easy to follow. Thank you.  

Line 89-90: This first sentence explaining the arm position, is a little confusing. Please rewrite it more clearly. "...in a posture..." is the confusing part. Seems to be a descriptor missing.

Line 100: It may be better to use the full term "Transcranial Magnetic Stimulation" for the subtitle.

Results are all clearly written and figures are very helpful for the reader. Although I think, perhaps Figure 4 would be better represented as a table. The legend is confusing and difficult to follow.

The discussion provides an interesting comparison with previous research and explains the results in the context of the findings from previous work. 

Round 2

Reviewer 1 Report

The authors answered my questions satisfactorily except for the implication of peripheral nerve stimulation on their outcome.

And I insist that statistics should review data analysis.

Sincerely

Reviewer 2 Report

The authors replied to all my comments and therfore The manuscript is suitable for publication. 
